# Electrophysiological Responses to Emotional Facial Expressions Following a Mild Traumatic Brain Injury

**DOI:** 10.3390/brainsci9060142

**Published:** 2019-06-18

**Authors:** Joanie Drapeau, Nathalie Gosselin, Isabelle Peretz, Michelle McKerral

**Affiliations:** 1Centre for Interdisciplinary Research in Rehabilitation (CRIR), IURDPM, CIUSSS du Centre-Sud-de-l’Île-de-Montréal, Montreal, QC H3S 2J4, Canada; joanie.drapeau@gmail.com; 2Departement of Psychology, Université de Montréal, Montreal, QC H3C 3J7, Canada; nathalie.gosselin@umontreal.ca (N.G.); isabelle.peretz@umontreal.ca (I.P.); 3International Laboratory for Brain, Music and Sound Research (BRAMS), Montreal, QC H3C 3J7, Canada

**Keywords:** mild traumatic brain injury, event related potentials, emotions, facial expressions

## Abstract

The present study aimed to measure neural information processing underlying emotional recognition from facial expressions in adults having sustained a mild traumatic brain injury (mTBI) as compared to healthy individuals. We thus measured early (N1, N170) and later (N2) event-related potential (ERP) components during presentation of fearful, neutral, and happy facial expressions in 10 adults with mTBI and 11 control participants. Findings indicated significant differences between groups, irrespective of emotional expression, in the early attentional stage (N1), which was altered in mTBI. The two groups showed similar perceptual integration of facial features (N170), with greater amplitude for fearful facial expressions in the right hemisphere. At a higher-level emotional discrimination stage (N2), both groups demonstrated preferential processing for fear as compared to happiness and neutrality. These findings suggest a reduced early selective attentional processing following mTBI, but no impact on the perceptual and higher-level cognitive processes stages. This study contributes to further improving our comprehension of attentional versus emotional recognition following a mild TBI.

## 1. Introduction

Traumatic brain injury (TBI) can lead to long-term physical, behavioral, and cognitive consequences affecting social participation [1,2]. A link between such difficulties and poor ability in recognizing emotions from facial expressions has also been suggested [3,4,5]. Indeed, emotional recognition is of crucial importance for successful social interactions. In particular, emotional facial expressions provide central information to understand others’ intentions and ultimately guide behavior. 

Deficits in recognizing emotions from facial expressions, especially negative emotions (i.e., fear, anger, sadness, or disgust) as compared to positive ones (i.e., happiness), have been established in prior behavioral studies following moderate to severe TBI [3,4,5,6,7,8,9,10,11]. In fact, brain regions involved in emotional processing, such as the prefrontal cortex (i.e., ventromedial and orbitofrontal) and limbic structures (i.e., amygdala, temporal lobes, fusiform gyrus) [12,13,14], are frequently damaged or disrupted after a moderate or severe TBI, and even following mild TBI (mTBI) [15,16]. Specific nontraumatic lesions to ventromedial prefrontal lobe, anteromedial temporal lobe, or bilateral amygdala have been related to deficits in discriminating fearful and disgusted facial expressions [17,18,19]. The electrophysiological literature on moderate and severe TBI in adults has also demonstrated dysfunction in cortical processing of emotional facial expressions [20,21].

Despite compelling evidence of structural changes and neurochemical alterations following mTBI that occur in brain regions associated with emotional processing [16,22,23,24], little is known about emotional recognition from facial expressions after mTBI. In a previous behavioral study [10], adults with uncomplicated mTBI (i.e., without intracranial abnormality) showed comparable performance to healthy controls when recognizing happy, sad, and fearful faces, which contrasted with the performance of individuals with a complicated mTBI (i.e., with intracranial abnormality) or with moderate or severe TBI, who showed significant impairment in recognition of fearful faces. Prior studies of cognitive functioning suggested that standard neuropsychological tests or behavioral methods often fail to detect dysfunctions in the postacute stages of mTBI, whereas more precise and sensitive measures, such as event-related potentials (ERPs), can identify subtle neurofunctional changes related to, for example, working memory and visual attention [25,26].

ERPs represent an ideal method to reveal the rapid temporal sequence of neural processing underlying emotional recognition [27,28], making it possible to distinguish the relative contributions of early (e.g., attentional and perceptual factors) and later processing stages (e.g., executive components of attention, discrimination of emotions). Several early and late ERP components have been studied in relation to emotional processing from facial expressions in healthy individuals, such as the N1 (early selective attentional processing) [29], N170 (perceptual integration of facial features) [30], and anterior N2 (executive cognitive control [31,32] or high-level cognitive processes enabling discrimination between emotional categories [33,34]).

Some studies have reported that these ERP components are also modulated by emotional valence (negative, neutral, positive) of facial expressions; for instance, preferential processing (i.e., increased ERP amplitude) for negative emotional facial expressions has been demonstrated, as compared to neutral facial expressions [27,33,34,35,36,37]. Especially, fearful stimuli are thought to be prioritized for attention because of their biological, evolutionary, and behavioral relevance [38]. Furthermore, it has been proposed that performing an explicit cognitive task, such as categorization or oddball discrimination, may suppress or interfere with the automatic preferential processing of emotional stimuli [39]. Thus, an implicit or passive viewing task could be more appropriate to reflect the effect of emotional valence.

To our knowledge, only two studies have explored electrophysiological responses specifically related to the perceptual integration of emotional facial expressions following mTBI (i.e., N170) [40,41]. In the study of Zuj et al. [40], previously deployed military personnel having sustained an mTBI showed, compared to a noninjured military group, a reduction in the N170 component for all facial expressions studied (i.e., fearful, angry, happy, and neutral). In D’Hont et al. [41], the authors showed, in preschool children having sustained an mTBI, an absence of preferential processing of emotional facial expressions in the early stages (P1, N170), as compared to controls.

The present study aimed to measure neural information processing underlying emotional recognition from facial expressions in adults having sustained an mTBI in comparison to healthy uninjured individuals. To do so, we conducted an experiment in which we measured early (N1, N170) and later (N2) ERP components during presentation of fearful, neutral, and happy facial expressions. Rare stimuli (butterflies), which were irrelevant to the study, were also presented to ensure that all participants paid attention to stimuli. We postulated that ERPs reflecting the early attentional processing stage would be affected following mTBI as compared to healthy controls. Also, since the cortical processing of emotions from facial expressions has been shown to be altered following an mTBI, we expected that participants with mTBI would also demonstrate ERP changes associated with perceptual integration and discrimination of emotional facial expressions. 

## 2. Materials and Methods

### 2.1. Participants

Thirty-nine participants were recruited for this study and the data of 21 participants were retained for primary analysis: 10 participants with uncomplicated (i.e., absence of intracranial lesions on brain imaging) mTBI and 11 healthy individuals. Nine participants with mTBI and five healthy individuals were removed from the analysis because of loss of trials (more than 50% of trials) due to eye blinks, movements, excessive sweating, or very large alpha oscillations which were mainly related to temporary temperature control issues in the testing environment. Number of trials included in EEG averaging were similar across conditions and groups, ranging from 35 to 50 trials per emotion category. Participants with TBI were recruited from the Centre de réadaptation Lucie-Bruneau in Montreal, Québec, Canada. Healthy controls were recruited through advertisements in the community. The study was approved by the Centre de recherche interdisciplinaire en réadaptation (CRIR) Research Ethics Committee. All participants provided a written informed consent prior to testing. 

Participants included in the study had to have sustained an mTBI during a road traffic accident or a fall, and have presented at least one of the following parameters, as noted in medical records: Glasgow Coma Scale (GCS) score (13–15/15), post-traumatic amnesia (PTA) duration (<24 h), and loss of consciousness (LOC) duration (<30 min) [42]. A negative CT scan or MRI result (no intracranial lesions) was also required (i.e., uncomplicated mTBI). Exclusion criteria for participants with TBI were: (1) younger than 18 or older than 55 years; (2) a psychiatric or neurological history, including more than one TBI; (3) a penetrating brain injury, such as assault with blunt or sharp object; (4) an uncorrected visual impairment; and (5) uses of psychostimulant, antidepressant, or antipsychotic treatment. This information was documented in the medical records or during the first appointment. An additional exclusion criterion for healthy controls was a history of TBI.

For group comparison purposes, intellectual functioning estimates (verbal IQ {vocabulary} and performance IQ {matrix reasoning} of the Wechsler adult intelligence scale-III {WAIS-III} [43], post-concussion symptoms (Post-Concussion Scale, PCS; 22 items rated 0-asymptomatic to 6-severely symptomatic, for a maximum score of 132 [44]) and symptoms related to depression (Beck Depression Inventory-II, BDI-II; 21 items rated 0-symptom absent to 3-severe symptoms, for a maximum score of 63 [45]) were measured. 

### 2.2. Stimuli

Thirty grayscale static facial expressions (256 × 256 pixels) were selected from the STOIC database (http://www.mapageweb.umontreal.ca/gosselif/sroyetal_sub.pdf) [46]. Facial expressions were created by professional actors (five men, five women) and were intended to express fear, happiness, and neutrality (10 different stimuli per emotion category). These facial stimuli were cropped to exclude nonfacial cues and were equivalent in mean luminance and contrast. A grayscale picture of a butterfly (256 × 256 pixels) was also included. Examples of stimuli are presented in Figure 1. 

### 2.3. Task and Procedure

Participants were presented with 330 trials in two blocks of 165 trials each (50 facial expressions per emotion category: fear, neutrality, happiness, and 15 butterflies), in a pseudo-random order, after a practice block of 33 trials (10 facial expressions per emotion category and three butterflies). All stimuli appeared at the center of the computer screen and were viewed from a distance of 1.14 meters. Each trial began with a fixation cross presented for 500 milliseconds (ms), followed by a visual stimulus (facial expression or butterfly) for 200 ms and a blank screen for 50 ms. An example of the task is presented in Figure 2. Participants were instructed to maintain fixation at the center of the computer screen, to avoid eye movements or blinks during visual stimuli presentation, and to press the space bar on the computer keyboard whenever a butterfly appeared. E-prime software was used for stimulus presentation and response recording. 

### 2.4. ERP Data Recording and Analysis

EEG activity was recorded while participants performed the task, with 64 Ag/AgCl active electrodes (Active Two BioSemi) placed over the scalp, according to the extended 10–20 International EEG system [47]. Horizontal eye movements (HEOG) were recorded with electrodes placed on the external canthus of each eye. Vertical eye movements and eye blinks (VEOG) were recorded from an electrode placed on the inferior orbital region of the left eye and Fp1 electrode. Sampling rate was set to 512 Hz and the data were referenced to the average mastoids during analysis. 

Brain Vision Analyzer (version 1.5) was used to analyze EEG data. Signals were filtered (0.01 Hz high-pass, 30 Hz low-pass) and were averaged offline. Eye movement artifacts were rejected semi-automatically with the ocular correction ICA method. Thus, trials with ocular artifacts at electrodes of interest (>70 µV), eye blinks (VEOG > 100 µV), and large horizontal eye movements (HEOG > 30 µV) were excluded from the analysis. EEGs were averaged separately for fear, neutrality, and happiness. For every trial, EEG epochs of 1000 ms (including a 200 ms prestimulus period) were averaged after artifact rejection. Epochs were then baseline corrected based on mean amplitude of activity recording during the 200 ms immediately prior to stimulus onset.

ERP components were measured (peak amplitude) by locating the highest peak within windows after stimulus onset [32]: N1 (50–150 ms), N170 (150–200 ms), and N2 (200–350 ms). According to prior emotion processing studies with healthy individuals, amplitude deflection of the N1 is largest over frontocentral electrodes [33,34], the N170 over lateral occipitotemporal sites [30,35,37], and the N2 at frontal sites [33,34].

### 2.5. Statistical Analysis

All ERP components were first investigated using visual inspection and amplitude measurement to detect which scalp regions (frontal, central, or posterior) and which electrode sites showed maximum amplitude. For each ERP component (N1, N170, N2) independently, data were analyzed using a general linear model repeated-measures ANOVA, with groups (two levels: mTBI, controls) as a between-subjects factor, and emotions (three levels: fear, neutrality, happiness) and electrode sites (when applicable) as within-subjects factors.

## 3. Results

### 3.1. Clinical and Demographic Data

Demographic and clinical characteristics of the participants are shown in Table 1. The mTBI group did not differ from healthy controls according to age, gender, years of education, and intellectual functioning estimates. Healthy controls and participants with mTBI differed significantly according to PCS score (F(1,19) = 10.675, *p* = 0.004) and BDI-II score (F(1,19) = 11.505, *p* = 0.003). All participants presented a BDI-II score of 20 or below (i.e., cut-off for the presence of a possible moderate–severe depression), except for two participants with mTBI. However, these two participants scored higher on ‘somatic’ and ‘cognitive’ items (e.g., fatigue, sleep disturbances, concentration difficulties) compared to ‘affective’ items (e.g., sadness), which could be more related to the mTBI than to depression. Indeed, BDI-II scores were highly correlated with post-concussion symptoms scores (*r* = 0.868, *p* = 0.000).

### 3.2. Behavioral Responses

All participants performed accurately on butterfly detection (healthy controls: mean 100%; mTBI group: mean 97.3%, standard deviation 3.1), ensuring that they all fully paid attention to the stimuli.

### 3.3. Electrophysiological Responses to Emotional Facial Expressions

#### 3.3.1. N1

N1 amplitude deflection was largest over FCz electrode (frontocentral site). Average waveforms within-subjects and average responses between-subjects for the mTBI group and the control group are shown in Figure 3, according to emotional expressions (fear, neutrality, happiness). 

Analysis revealed a significant main effect of group (F(1,19) = 4.774, *p* = 0.042, η^2^ = 0.201), indicating that adults with mTBI showed reduced N1 amplitude as compared to healthy individuals in processing emotional facial expressions. No significant main effect of emotion (*p* = 0.470) and no significant interaction between groups and emotions (*p* = 0.855) were found.

#### 3.3.2. N170 

N170 amplitude deflections were largest over P9 (left hemisphere) and P10 (right hemisphere) electrodes (occipitotemporal sites). Average waveforms within-subjects and average responses between-subjects for the mTBI group and the control group are shown in Figure 4, according to emotional expressions (fear, neutrality, happiness).

Analysis comparing mTBI and control groups revealed no significant main effect of group (*p* = 0.331) and no significant interaction between emotion and group (*p* = 0.205), but a significant main effect of emotion (F(2,38) = 5.818, *p* = 0.006, η^2^ = 0.234), indicating that some emotions elicited larger amplitude than others, irrespective of the group. Bonferroni-adjusted pairwise comparisons revealed that fearful facial expressions elicited larger amplitude than neutral ones (*p* = 0.007), as did happy facial expressions (*p* = 0.007). There was also a significant main effect of electrode site (F(1,19) = 7.090, *p* = 0.015, η^2^ = 0.272), revealing a stronger amplitude in the right hemisphere, tempered by a significant interaction between electrode site and emotion (F(2,38) = 4.720, *p* = 0.022, η^2^ = 0.344). Bonferroni-adjusted pairwise comparisons revealed that, specifically, fearful facial expressions showed larger amplitude in the right hemisphere (*p* = 0.01) as compared to other emotions. No significant interaction between electrode site and group (*p* = 0.513) and no interaction between electrode site, group, and emotion (*p* = 0.240) were found.

#### 3.3.3. N2

N2 amplitude deflections were largest over Fz electrode (frontal site). Average waveforms within-subjects and average responses between-subjects for the mTBI group and the control group are shown in Figure 5, according to emotional expressions (fear, neutrality, happiness). 

Analysis revealed no significant main effect of group (*p* = 0.346) nor significant interaction between emotion and group (*p* = 0. 112), but a significant main effect of emotion, F(1,19) = 9.452, *p* = 0.002, η^2^ = 0.512, indicating that some emotions elicited larger amplitude than others, irrespective of the group. Bonferroni-adjusted pairwise comparisons revealed that both groups showed a preferential processing (i.e., larger amplitude) for fearful facial expressions, as compared to neutrality (*p* = 0.011) and happiness (*p* = 0.002). 

## 4. Discussion

The aim of this original study was to measure neural information processing underlying emotion recognition from negative (fear), neutral, and positive (happy) facial expressions in adults having sustained an mTBI as compared to healthy individuals. According to our hypothesis, results indicated significant differences between groups, irrespective of emotional expressions, in the early attentional stage (N1), suggesting altered selective attentional processing for all emotions from facial expressions following mTBI, even in the context of attentional facilitation induced by butterfly detection. Findings also revealed that, contrary to what was hypothesized, the two groups showed similar perceptual integration of facial features (N170), with a preferential processing for fearful and happy facial expressions, as compared to neutral faces, which was stronger in the right hemisphere. Moreover, at a higher-level emotion discrimination stage (N2), both groups demonstrated preferential processing for fear as compared to happiness and neutrality.

Globally, these findings suggest modified early selective attentional processing following mTBI, in the context of preserved perceptual integration and higher-level emotion discrimination. Even though the sample size was small, the demonstrated effects were robust, with most effect sizes being strong, the others moderate. In a previous behavioral study, Drapeau et al. [10] also showed comparable performance between individuals with uncomplicated mTBI and healthy individuals for perceptual integration of facial features and facial emotion discrimination. These findings are also in line with previous research on mTBI or on repeated concussions which has previously documented such dysfunction in selective attention in general and specifically for emotional facial expressions [48,49,50].

In contrast, another study showed no attentional biases or enhanced processing of angry or fearful facial expressions and reduced N170 amplitudes to angry, fearful, and happy emotional facial expressions in a group of military personnel who had sustained an mTBI, compared to a non-TBI military group [41]. It is possible that in this latter study, the mTBIs sustained during deployment and combat reflected injuries toward the more severe spectrum of mTBI (similar to that of complicated mTBI, although intracranial injuries were not documented) due to the biomechanical forces involved in such contexts and since almost 40% of their sample had a previous mTBI, while this variable was an exclusion criterion in the present study. Hence, it is possible that individuals with mTBI who present more severe post-mTBI neuropathophysiological changes can show altered neural processing at higher-level cognitive stages enabling discrimination of fearful facial expressions, while adults having sustained a single uncomplicated mTBI appear to show changes only for selective attention, as demonstrated in the present study.

This is not surprising, as the level of cognitive functioning is associated with brain injury severity even in the mild severity spectrum [51,52]. In such cases, fearful or angry expressions in particular may overload attentional resources, thus negatively affecting subsequent facial expression processing as well as discrimination between emotion categories. Indeed, fear involves more facial features (e.g., raised forehead and eyebrows, open mouth, open eyes) to pay attention and to process, as compared to neutrality or happiness (i.e., smile). Some studies have also demonstrated that even in healthy individuals, attentional load can reduce facilitated processing of threatening faces during more elaborate cognitive stages involving executive attentional control [53] and also reduce activation of brain areas associated with fear processing, such as fusiform gyri [54]. 

Our findings of reduced early selective attentional processing following mTBI, but no impact, on the perceptual and higher-level cognitive processes stages related to facial emotion recognition also contrast with previous work showing impairments in negative emotion, especially fearful, recognition from facial expressions following moderate or severe TBI [3,5,6,8,9,10,11]. They also differ from prior studies indicating that altered emotion recognition from facial expressions is closely linked with attentional difficulties after moderate–severe TBI [55,56]. This study contributes to further improving our comprehension of attentional versus emotion recognition following an mTBI and points to the importance of carefully documenting the severity of TBI, even on the mild spectrum (e.g., according to common diagnostic criteria), when studying the different processing stages associated with recognition of facial emotional expressions, as different types and range of impairments may present according to TBI severity. For individuals with mTBI, interventions geared toward improving attentional functioning should be encouraged in individuals having sustained an mTBI and who show persistent post-mTBI symptoms after three months, as did the participants in the present study [57,58]. Training complex forms of attention could result in more effective allocation of attentional resources in general [59], although it is not known how this would specifically impact early attentional processing of emotional facial expressions.

## 5. Limitations

Sample size being a limitation of the present study, ERP studies in larger mTBI subgroups (e.g., uncomplicated, complicated, multiple mTBIs) are needed to better characterize the mechanisms underlying attentional processing and emotion recognition from facial expressions on the entire severity spectrum of mTBI. This would be of interest to identify those individuals who could benefit from cognitive intervention and remediation strategies. Future studies should also combine neuropsychological and behavioral measures with advanced high-definition functional brain imaging techniques (e.g., multiarray EEG and magnetoencephalography) in order to refine our understanding of the cortical underpinnings of normal versus altered attentional and facial expression processing after an mTBI.

## 6. Conclusions 

This original study is the first systematic exploration of the temporal sequence of neural processing underlying emotion recognition from facial expressions in adults having sustained a single uncomplicated mTBI as compared to healthy individuals. The present findings indicate affected early attentional processing following mTBI but no impact on perceptual and higher-level cognitive processing stages. This study contributes to further improving our comprehension of emotion recognition following TBI.

## Figures and Tables

**Figure 1 brainsci-09-00142-f001:**
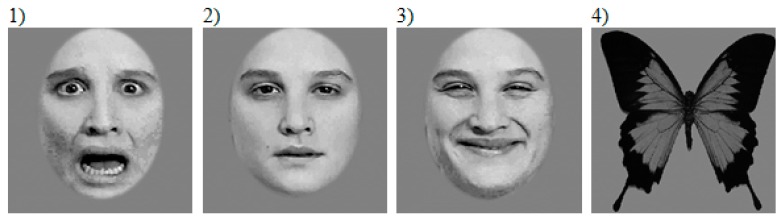
Examples of visual stimuli; (**1**) fearful facial expression, (**2**) neutral facial expression, (**3**) happy facial expression, (**4**) butterfly.

**Figure 2 brainsci-09-00142-f002:**
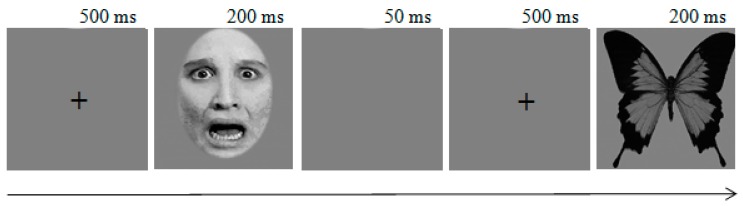
Example of the task.

**Figure 3 brainsci-09-00142-f003:**
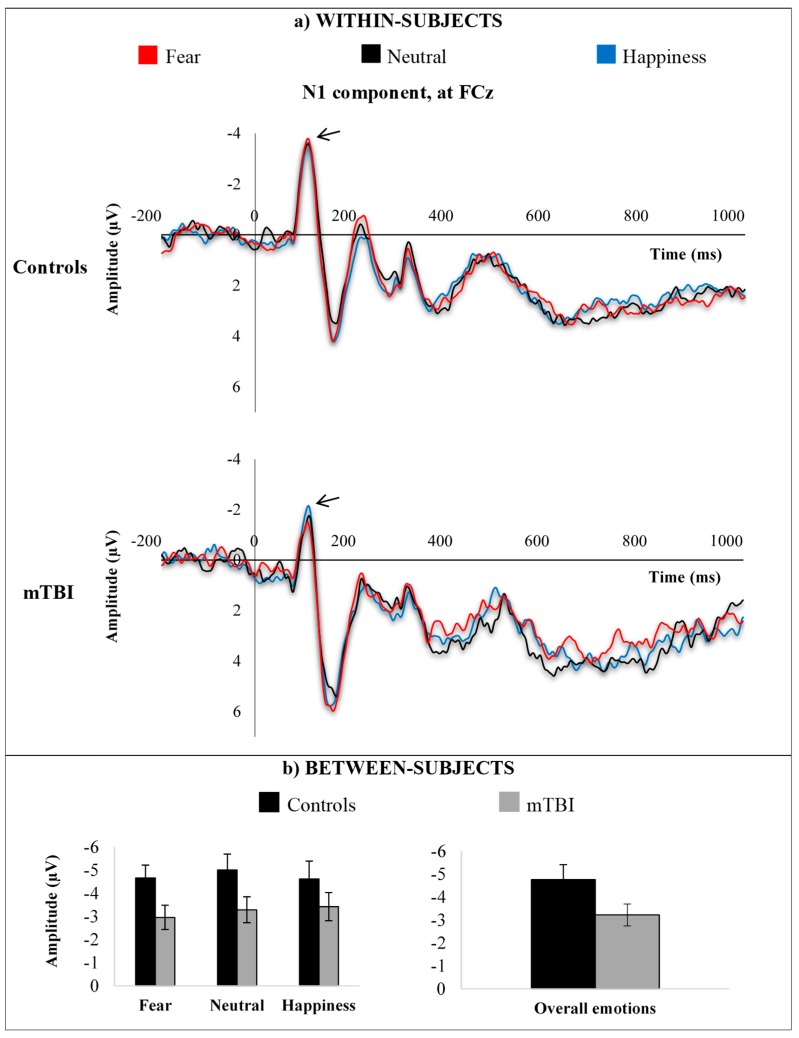
N1 responses at FCz electrode site, (**a**) within-subjects and (**b**) between-subjects, according to emotional expressions (fear, neutrality, happiness).

**Figure 4 brainsci-09-00142-f004:**
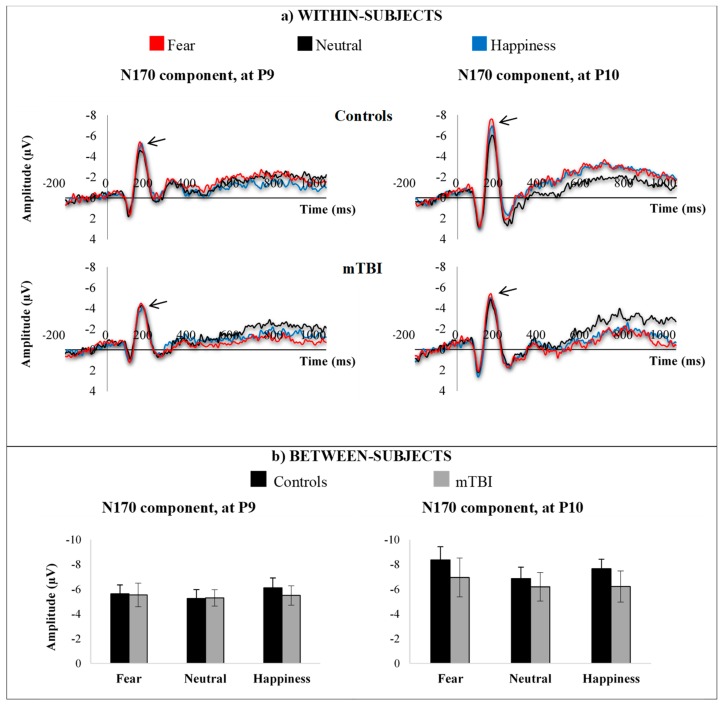
N170 responses at P9 electrode site (left hemisphere) and P10 electrode site (right hemisphere), (**a**) within-subjects and (**b**) between-subjects, according to emotional expressions (fear, neutrality, happiness).

**Figure 5 brainsci-09-00142-f005:**
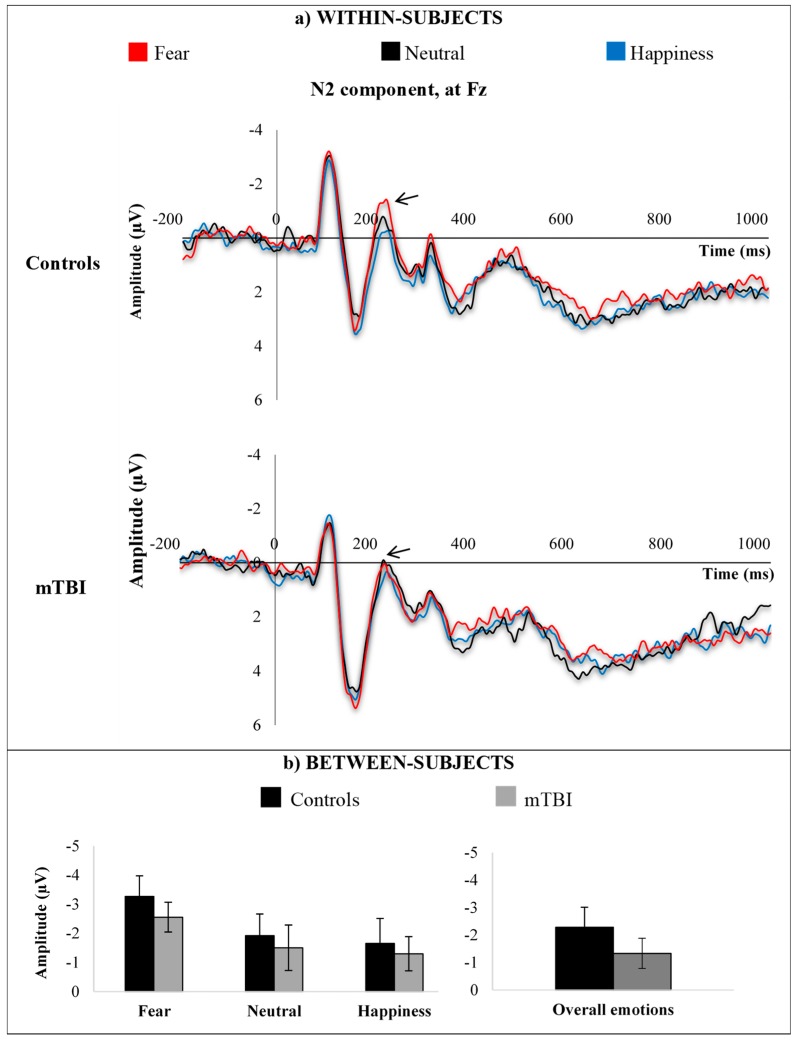
N2 responses at Fz electrodes site, (**a**) within-subjects and (**b**) between-subjects, according to emotional expressions (fear, neutrality, happiness).

**Table 1 brainsci-09-00142-t001:** Demographic and clinical characteristics of the mTBI and control participants.

TBI Participants	Sex	Age (years)	Education (years)	Cause of TBI	GCSScore	Lesion Site on CT Scan	Months Post-Injury	BDI-II	PCS	Verbal IQ Estimate	Performance IQ Estimate
mTBI16	F	26	13	Fall on ice	15	Normal	31	11	31	105	120
mTBI27	M	36	14	Fall >1-meter, work accident	15	Normal	28	6	4	95	95
mTBI30	F	33	18	Face-to-face hit, hockey	15	Normal	18	0	7	105	120
mTBI34	F	20	13	Car accident	15	Normal	4	20	26	100	115
mTBI41	F	46	17	Car accident	14	Normal	5	9	16	105	100
mTBI42	F	32	18	Knee-to-head hit, basketball	15	Normal	24	15	28	100	100
mTBI44	F	29	16	Fall backward sitting on bench	15	Normal	7	7	18	110	125
mTBI46	F	42	18	Hit on head by basketball	15	Normal	27	4	20	100	110
mTBI47	F	40	17	Car accident	15	Normal	6	27	40	95	120
mTBI48	F	42	16	Fall on head, martial arts	14	Normal	26	26	64	95	95
CTRL101	F	44	13					2	6	105	115
CTRL103	F	29	18					6	14	105	115
CTRL105	M	31	14					0	0	95	120
CTRL106	M	26	16					3	4	115	115
CTRL107	M	26	16					0	23	100	125
CTRL108	F	51	14					8	15	105	115
CTRL109	M	35	16					6	3	100	125
CTRL112	M	29	12					3	4	100	115
CTRL114	F	30	18					0	1	115	105
CTRL115	F	30	20					0	1	105	120
CTRL116	M	28	17					0	3	120	135

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
