# Peer review of "Electrophysiological Responses to Emotional Facial Expressions Following a Mild Traumatic Brain Injury"

_brainsci, 2019, doi:10.3390/brainsci9060142_

Reviewer 1 Report

The Authors have considerably improved the paper.

One more observation:
Line 67: ERP by definition cannot reveal “deficits” (i.e., “a lack or impairment in an ability or functional capacity”, Merriam Webster Dictionary), they can reveal changes/modifications/differences in neural processing.

The same for instance at line 303, 310 and 335, Where the terms "reduced" or "decreased" seem to imply cognitive deficits; the sentences should read, more appropriately, "altered" or "modified" processing.

Minor point: line 305: "both groups" should read "the two goups"

Author Response

Response to comments from Reviewer 1

"Line 67: ERP by definition cannot reveal “deficits” (i.e., “a lack or impairment in an ability or functional capacity”, Merriam Webster Dictionary), they can reveal changes/modifications/differences in neural processing.

The same for instance at line 303, 310 and 335, Where the terms "reduced" or "decreased" seem to imply cognitive deficits; the sentences should read, more appropriately, "altered" or "modified" processing."

The requested changes have been made to the revised maniscript at lines 67, 304, 311, and 337.

"Minor point: line 305: "both groups" should read "the two goups""

The requested change has been made at line 306.

Reviewer 2 Report

The authors  report their comparison of electro-cortical responses for emotional perception in patient with minor traumatic brain injury compared to healthy controls. Those with minor traumatic brain injury had findings consistent with early selective attentional processing. The study is well performed with clear differences evident between the groups

One suggested change would be in the last sentence of the last paragraph in the introduction. In that secondary aim, the authors should more clearly indicate the cortical changes expected are those identified on the basis of electroencephalographic information.

Author Response

Response to comments from Reviewer 2

"One suggested change would be in the last sentence of the last paragraph in the introduction. In that secondary aim, the authors should more clearly indicate the cortical changes expected are those identified on the basis of electroencephalographic information."

The requested change has been made at lines 116 and 123.

This manuscript is a resubmission of an earlier submission. The following is a list of the peer review reports and author responses from that submission.

Round  1

Reviewer 1 Report

The Authors explore possible ERP alterations in mild TBI (mTBI) patients during emotional face expression observation; they find significant differences between mTBI and controls only in early attentional stages (N1)

My major criticism are:

1) It is never explained what justifies grouping the moderate and severe with the mild TBI patients. If there are data in the literature supporting this, it is not clear why the Authors do not mention them and so why they did not group all the patients from the beginning. Otherwise, the Authors should verify whether the three groups are statistically different, and only in case they are not, they could group them; however, having only 2 moderate + 2 severe TBI patients, it is not reasonable to do statistics here.

Furthermore, a relevant part of the discussion (lines 308 and following) is based on the results obtained from the whole group including the moderate and severe patients, confronting these results with those obtained from just the mTBI, as if the Authors had actually compared mild to moderate-severe TBI patients, which they did not (and could not do, with these small numbers).

Also, classification of TBI criteria are defined for mTBI, but not as a general framework and specifically not for moderate and severe TBI.

2) The paper ingenerates a constant confusion about presence or absence of deficits in emotion recognition; it appears that the Authors started this research in order to explain behavioral deficits, which in reality have not been demonstrated, neither in this paper, nor in the literature.

For instance, at lines 69 and following, which deficits are the Authors referring to in the sentence “these emotional recognition from facial expressions deficits”? This sentence contradicts the previous paragraph stating the absence of behavioral deficits in mild TMI.

Also, at lines 297 and following, “These novel ERP results are consistent with a previous behavioral study (Drapeau, Gosselin, Peretz & McKerral, 2017)” is not warranted by the results, which point to no differences between controls and mTBI.

Other points:

i) Instructions for Authors for this journal require that “References must be numbered in order of appearance in the text”

ii) line 18 and line 290: “Both groups” should be “The two groups”

iii) lines 48 and following: the Authors should specify that lesions referred to in this sentence are non-traumatic

iv) line 88: “suppress or interfere”

v) lines 391-392: why are some author names underlined?

vi) line 396: reference “Drapeau, J., Gosselin, N., Peretz, I., and McKerral, M. (accepted)” has actually been published 2 years ago!

vi) Figures 3-4-5: some small-font captions are blurred and almost impossible to read

Author Response

Reviewer 1

The Authors explore possible ERP alterations in mild TBI (mTBI) patients during emotional face expression observation; they find significant differences between mTBI and controls only in early attentional stages (N1).

My major criticism are:

1) It is never explained what justifies grouping the moderate and severe with the mild TBI patients. If there are data in the literature supporting this, it is not clear why the Authors do not mention them and so why they did not group all the patients from the beginning. Otherwise, the Authors should verify whether the three groups are statistically different, and only in case they are not, they could group them; however, having only 2 moderate + 2 severe TBI patients, it is not reasonable to do statistics here. Furthermore, a relevant part of the discussion (lines 308 and following) is based on the results obtained from the whole group including the moderate and severe patients, confronting these results with those obtained from just the mTBI, as if the Authors had actually compared mild to moderate-severe TBI patients, which they did not (and could not do, with these small numbers). Also, classification of TBI criteria are defined for mTBI, but not as a general framework and specifically not for moderate and severe TBI.

Response: We thank the reviewer for their highly pertinent and constructive comments. To respond to this reviewer’s criticism, in the revised manuscript we have included only participants with mTBI and uninjured control participants. The introduction, results (including new figures) and discussion sections have been considerably rewritten in this light and, in our view, this increases the quality and coherence of the paper.

2) The paper ingenerates a constant confusion about presence or absence of deficits in emotion recognition; it appears that the Authors started this research in order to explain behavioral deficits, which in reality have not been demonstrated, neither in this paper, nor in the literature. For instance, at lines 69 and following, which deficits are the Authors referring to in the sentence “these emotional recognition from facial expressions deficits”? This sentence contradicts the previous paragraph stating the absence of behavioral deficits in mild TMI. Also, at lines 297 and following, “These novel ERP results are consistent with a previous behavioral study (Drapeau, Gosselin, Peretz & McKerral, 2017)” is not warranted by the results, which point to no differences between controls and mTBI.

Response: This confusion has been addressed in the various sections of the revised manuscript. Only early attentional deficits were identified in the mTBI group, but no effect was found for perceptual and higher-level cognitive processing stages, and this has been more clearly presented and discussed as so in the revised manuscript.

Other points:

i) Instructions for Authors for this journal require that “References must be numbered in order of appearance in the text”

Response: In the revised manuscript, references are now numbered and in brackets in the text.

ii) line 18 and line 290: “Both groups” should be “The two groups”

Response: This was corrected, as suggested. P a g e 2 | 3

iii) lines 48 and following: the Authors should specify that lesions referred to in this sentence are non-traumatic

Response: This was done, as suggested.

iv) line 88: “suppress or interfere”

Response: This was corrected, as suggested.

v) lines 391-392: why are some author names underlined?

Response: The underlining was a formatting error and was removed.

vi) line 396: reference “Drapeau, J., Gosselin, N., Peretz, I., and McKerral, M. (accepted)” has actually been published 2 years ago!

Response: The reference was updated, as suggested.

vi) Figures 3-4-5: some small-font captions are blurred and almost impossible to read

Response: The figures were redone to show only results from participants with mTBI and control participants, and are now of better quality.

Reviewer 2 Report

Overall, I believe the study offers an important incremental result regarding the neuropsychological underpinnings of emotional processing following traumatic brain injury. The study is well designed and is clearly presented. I have no major concerns and only a few minor points outlined below:

Title: the title is somewhat inaccurate in regard to mild traumatic brain injury as both results and discussions focuses on comparing the findings in the mTBI and TBI groups (In particular, all figures in the results section are on the overall group). Perhaps the authors may want to revise it to reflect the overall approach they took in the data analysis.

Methods:

The number of participants not included in the analysis is quite puzzling and atypical. with 14 of them removed out of 39, that is a 35% exclusion rate.

Given that participants will less than 50% of good trials were rejected, which is already quite a low threshold, I believe the authors should indicate the average number of trials per conditions in the remaining pool of participants. More importantly, given that the number of trials included to compute the ERPs can influence the signal-to-noise ratio, and thus the probability to detect an effect, It would be particularly pertinent to indicate how the number of trials per condition is comparable across groups.

The authors should clarify if the ANOVAs were conducted on mean amplitude in the predetermined latency bands or on the peaks at the chosen electrode sites.

The authors should also indicate how they chose the electrode sites for each component. Was it based only on visual inspection? 

Line 194: given that the authors did not perform source analysis, they cannot infer brain regions. It thus seems more appropriate to says "scalp region" than "brain region" in this context.

Author Response

Reviewer 2

Overall, I believe the study offers an important incremental result regarding the neuropsychological underpinnings of emotional processing following traumatic brain injury. The study is well designed and is clearly presented. I have no major concerns and only a few minor points outlined below:

Title: the title is somewhat inaccurate in regard to mild traumatic brain injury as both results and discussions focuses on comparing the findings in the mTBI and TBI groups (In particular, all figures in the results section are on the overall group). Perhaps the authors may want to revise it to reflect the overall approach they took in the data analysis.

Response: We thank the reviewer for their insightful and constructive comments. In view of the comments made by Reviewer 1 and those made by this reviewer, and to increase the quality and coherence of the paper, in the revised manuscript we chose to include only participants with mTBI and uninjured control participants. The introduction, results (including new figures) and discussion sections have been considerably rewritten in this light. The title of the manuscript thus remains the same and now reflects well the content of the revised paper.

Methods:

The number of participants not included in the analysis is quite puzzling and atypical. with 14 of them removed out of 39, that is a 35% exclusion rate. Given that participants will less than 50% of good trials were rejected, which is already quite a low threshold, I believe the authors should indicate the average number of trials per conditions in the remaining pool of participants. More importantly, given that the number of trials included to compute the ERPs can influence the signal-to-noise ratio, and thus the probability to detect an effect, It would be particularly pertinent to indicate how the number of trials per condition is comparable across groups.

Response: Information was added on number of averaged trials per condition and the trial rejection rate was better explained. P a g e 3 | 3

The authors should clarify if the ANOVAs were conducted on mean amplitude in the predetermined latency bands or on the peaks at the chosen electrode sites.

Response: The revised manuscript now indicates that peak amplitudes at chosen electrode sites were used for statistical analyses.

The authors should also indicate how they chose the electrode sites for each component. Was it based only on visual inspection?

Response: In the methods section of the revised manuscript it has been added that ERP components were determined using visual inspection and amplitude measurement to determine at which electrode sites amplitudes were maximal.

Line 194: given that the authors did not perform source analysis, they cannot infer brain regions. It thus seems more appropriate to says "scalp region" than "brain region" in this context.

Response: This was corrected, as suggested.